# Tracking momentary attention fluctuations with an EEG-based cognitive brain-machine interface

## Abstract

Momentary fluctuations in attention (perceptual accuracy) correlate with neural activity fluctuations in primate visual areas. Yet, the link between such momentary neural fluctuations and attention state remains to be shown in the human brain. We investigate this link using a real-time cognitive brain machine interface (cBMI) based on steady state visually evoked potentials (SSVEPs): occipital EEG potentials evoked by rhythmically flashing stimuli. Tracking momentary fluctuations in SSVEP power, in real-time, we presented stimuli time-locked to when this power reached (predetermined) high or low thresholds. We observed a significant increase in discrimination accuracy ($d'$) when stimuli were triggered during high (versus low) SSVEP power epochs, at the location cued for attention. Our results indicate a direct link between attention's effects on perceptual accuracy and and neural gain in EEG-SSVEP power, in the human brain.

## 1 Introduction

Visual attention improves perceptual accuracy and reaction times for attended stimuli, and also enhances the activity (gain) of visual neurons. Previous studies have shown that momentary fluctuations in attention can be measured by tracking neural activity in primate visual areas (e.g. V4) (2, 7); such fluctuations correlate with animals' perceptual accuracy on a trial-by-trial basis. Yet, the link between such momentary neural fluctuations and attention state remains to be shown in the human brain.

Here, we investigated this link using a real-time cognitive brain machine interface (Fig. 1A), based on steady state visually evoked potentials (SSVEPs): occipital EEG potentials evoked by periodically flickering stimuli, whose power systematically modulates with attention (6). Our results show that EEG-SSVEP power can be used to track attentional fluctuations, in real-time, in the human brain.

## 2 EEG cBMI system

**Real-time system:** The cBMI system broadly comprises the presentation system, the EEG acquisition system and the processing system (Fig. 1A). The presentation system (Intel Core i5, 8 GB RAM, Win 7) was used to present accurately controlled visual stimuli on the monitor (screen refresh rate = 144 Hz) and send task specific events to the EEG acquisition system using a parallel port. Stimuli were presented with Psychtoolbox. For EEG acquisition, we used the Biosemi 128-channel ActiTwo System for EEG acquisition. The acquisition system synchronized the task specific events with EEG data, and sent it to the processing system. The processing system (Intel Core i7, 16 GB RAM, Win 10) acquired the EEG and events data packet using an acquisition software. The data was processed and neurofeedback was generated. The neurofeedback was then sent to the presentation system by means of a shared file accessible by both systems.

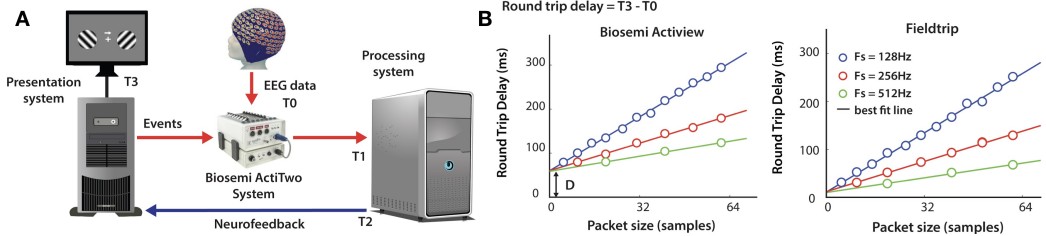

Figure 1: **Real-time tracking of attention fluctuations with a cBMI system. (A)** Schematic of the real-time cBMI system. The system acquires EEG signals in real-time, processes it and delivers a neurofeedback with closed-loop delays of the order of 10 ms. **(B)** Round trip (closed-loop) delay using Biosemi ActiView (left) and Fieldtrip (right) acquisition software (overhead: D, offset).

To identify efficient acquisition software, we compared the round-trip (closed-loop) delay for four acquisition software: ActiView, Lab Streaming Layer, OpenVibe and Fieldtrip (Fig. 1B). We measured round-trip delay by varying the EEG+event packet size, across different sampling frequencies, fit a line to the data, and estimated the intercept, which is a measure of the overhead. We observed that Fieldtrip produced the least overhead of $10.98 \pm 0.50$ ms.

**EEG data recording:** Scalp EEG recordings were performed with 41 occipital electrodes out of the total 128 electrodes. The data was streamed in real-time using the Fieldtrip buffer at 128 Hz. EEG data was also stored at 4096 Hz for offline analyses. Spectral analysis was performed using Chronux 2.12 toolbox (1) EEGLAB 13.6.5b (4) functions were used to generate the topographical plots. EEG data preprocessing followed standard protocols (SCADS: Statistical Correction of Artifacts in Dense-array Studies (5)). Finally, the EEG data was re-referenced to the average reference.

## 3   Experimental task design

Participants (n=15) performed the task in a dark, sound-attenuated room. Experimental protocols were approved by the Institute Human Ethics Committee at IISc. The participant's head was positioned 60 cm away from the monitor on a chin rest. The task was presented on a 24 inch LED monitor and a resolution of 1920 by 1080 at 144 Hz of screen refresh rate. We used MATLAB (Mathworks Inc.) based Psychtoolbox for the psychophysical task design. A Cedrus response box (RB-540) was used to record responses.

The experiment began when a white fixation cross was presented in the center of a gray screen (Fig. 2A). Subjects were instructed to fixate on the cross throughout a trial. Two flickering stimuli (pedestals) appeared, each produced by superimposing gratings oriented at $45°$ and $-45°$ from the horizontal. The pedestals were presented on either side of the fixation cross. Each pedestal flickered at a distinct frequency to evoke EEG oscillations at the corresponding frequency, known as Steady State Visually Evoked Potential or SSVEPs.

After 1000 ms, a directed cue (central arrow) appeared, indicating the side to be attended. The cue was 100% valid and counterbalanced across left and right stimuli. Some time later (depending on SSVEP power, see next section), the pedestals disappeared, and task stimuli (oriented gratings at $\pm45°$) on each side) appeared briefly for 75 ms. Subjects detected and reported the orientation of the grating on the cued side (Target), ignoring the grating on the uncued side (Distractor). We used signal detection theory (Fig. 2A, inset) to measure the discriminability $d'$ and decision bias ($c$) for discriminating clockwise from counterclockwise target stimuli.

## 4   Isolating SSVEP components and real-time triggering

We employed a dimensionality reduction algorithm, Denoising Source Separation (DSS) to isolate and quantify, with high SNR (signal-to-noise ratio), SSVEP power evoked by the flickering pedestals (3). Briefly, DSS identifies low-dimensional sources from high-dimensional, noisy sensor data. It involves whitening of the considered sensor signals, filtering for the desired feature (here power at a particular SSVEP frequency), followed by rotating the data along a direction that maximizes the

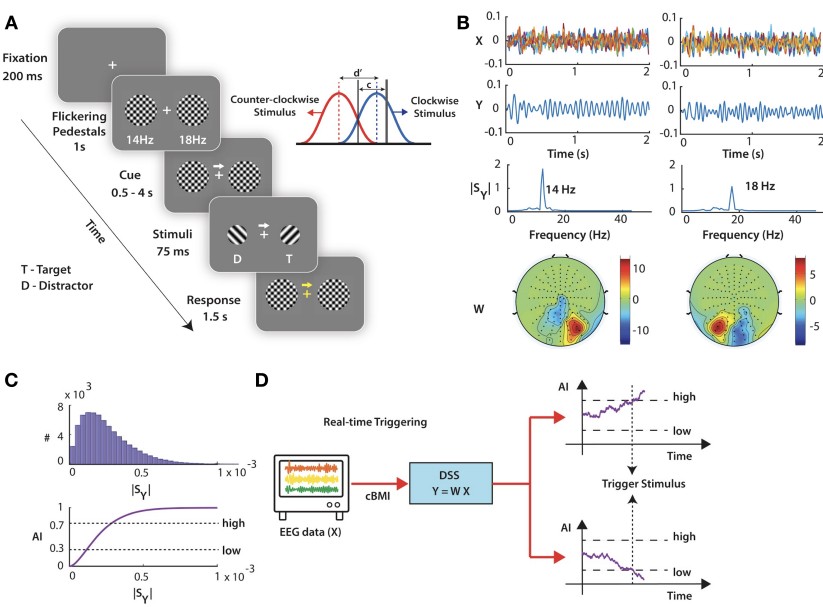

Figure 2: **Schematic of real-time triggering.** **(A)** Attention task timeline. **(B)** Dimensionality reduction and SSVEP isolation using Denoising Source Separation. **(C)** (top) SSVEP distributions from a baseline block. (bottom) Thresholds estimated using the CDF of this distribution (Attention idex or AI). **(D)** Schematic of task stimulus triggering based on high and low thresholds for the AI.

variance along the desired feature. In our case, each DSS latent dimension ($Y$) is a linear combination ($Y = WX$) of the raw EEG signals ($X$) from the occipital electrodes. For these analyses, we identified, with visual inspection, the DSS dimension with the clearest peak in the spectrum at the corresponding SSVEP frequency (Fig. 2B); typically, this was the first DSS dimension.

Next, we employed the following procedure for triggering task stimuli (the grating discriminanda): We conducted a "baseline" block before the actual experiment. In the baseline block, no attention cues were presented and the duration between the onset of the flickering pedestals and the task stimuli were selected randomly from an exponential distribution, with a minimum duration of 2.5 s and a maximum of 5 s. We estimated SSVEP power in moving windows of 0.5 s (64 samples) each with a step-size of 7.8 ms (1 sample), using multi-taper spectral analyses using one Slepian taper, to generate an SSVEP power distribution across time for each SSVEP frequency (Fig. 2C, top). Next, we fit a non-parametric distribution to this distribution, and calculated the cumulative distribution function (CDF). This CDF provides a normalized measure of SSVEP power, which accounts for variations in baseline SSVEP power across subjects. The CDF value corresponding to the SSVEP power evoked by the pedestal on each side (cued/target, or uncued/distractor) was termed the attention index (AI) for that side (Fig. 2C, bottom).

In the actual experiment, we tracked, in real time the AI of the pedestal the cued (target) side or uncued (distractor) side, on separate trials. When the AI on the respective side reached a particular "high" or "low" threshold value, the presentation of the task stimuli (gratings) was immediately triggered (Fig. 2D). This paradigm enabled us to measure a direct link between the participants' behavioral accuracy when the EEG-SSVEP power was in particular states (high or low) at the target or distractor locations, immediately preceding their presentation.

## 5   Results

All 15 participants were tested on trials with target-based triggering: task stimulus (grating) presentation was triggered when the attention index (AI) based on the SSVEP power of the cued (target) flickering pedestal crossed a high (AI-high) or low (AI-low) threshold. We tested for behavioral differences between the AI-high and AI-low conditions. Fig. 3A shows the comparison of four behavioral measures: percentage correct, reaction time, discrimination accuracy ($d'$) and choice

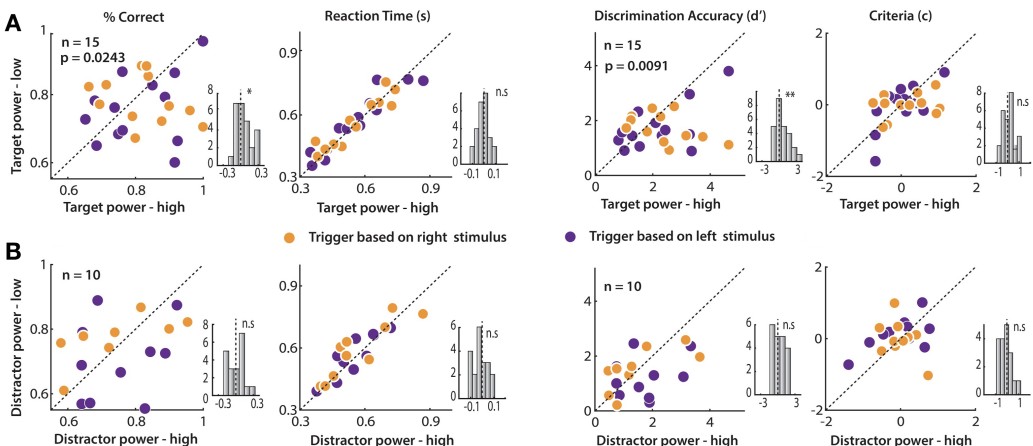

Figure 3: **Behavioral measures of attention based on SSVEP power. (A)** Scatter plots of the values of four behavioral metrics, for the target SSVEP triggered condition. x-axis: AI-high trials; y-axis: AI-low trials. **(B)** Same as in (A), but for the distractor SSVEP triggered condition.

criterion ($c$) for AI-high vs AI-low trials. Percentage correct was marginally higher for AI-high as compared to AI-low trials (n-way ANOVA p = 0.026). Discrimination accuracy ($d'$) was significantly higher, across the population, for AI-high trials as compared to AI-low trials (p < 0.01). We did not observe any significant differences in other measures like reaction time and criterion (p > 0.3).

In a subset of the participants (n=10), we also triggered task stimuli based on the uncued side (distractor) SSVEP power (AI). Fig. 3B shows the comparision of the same four behavioral measures: percentage correct, reaction time, $d'$ and choice criterion ($c$) for distractor based AI-high vs AI-low trials. We observed no systematic effects for any of the four behavioral measures (ANOVA p > 0.3) with distractor based triggering. In summary, discrimination accuracy, $d'$, was higher when the SSVEP power on the target (cued) side was high, compared to when it was low. Furthermore, the $d'$ effect did not depend on distractor SSVEP power. Neither target- nor distractor-based triggering produced reliable effects on reaction times.

Overall, our results indicate a direct link between attentional effects on perceptual accuracy and neural gain in EEG-SSVEP power, in the human brain. Furthermore, the results suggest that neural mechanisms that mediate target enhancement may be distinct from those that mediate distractor suppression. Additionally, attention's effects on behavioral accuracies and reaction times may engage distinct neural mechanisms.

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
