# OpenReview forum: "Tracking momentary attention fluctuations with an EEG-based cognitive brain-machine interface"
_NeurIPS.cc/2019/Workshop/Neuro_AI — Submitted to Real Neurons & Hidden Units @ NeurIPS 2019_

### Official Review · AnonReviewer1 · 2019-09-22
**Possibly interesting application, but the design and the analysis are not conclusive**

**Clarity:** 2

**Comment:**

A very elegant and comprehensive study on the effect of attention on the amplitude of resonance frequencies is the following one

Gulbinaite, R., Roozendaal, D. H. M., & VanRullen, R. (2019). Attention differentially modulates the amplitude of resonance frequencies in the visual cortex. NeuroImage, 203, 116146. doi:10.1016/j.neuroimage.2019.116146

Of course it is not the sole way of addressing these issue, but most of the points that seemed not clear to me in this submission, are properly addressed in the paper above.

**Category:**

Not applicable

**Clarity Comment:**

The frequency of the flickering stimuly is never mentioned. So I had to guess that it was 14 and 18 Hz and trusted the authors on their correct isolation.

The choice of ANOVA tests is not clear.

**Evaluation:**

2: Poor

**Importance:**

2: Marginally important

**Importance Comment:**

The paper is very ambitious. If all the claims in the conclusions could be verified, this could result in an important contribution, but there's no evidence for any of them.

**Intersection:**

2: Low

**Intersection Comment:**

Not so much AI.

**Rigor Comment:**

It is not clear why the amplitude was binarized in high versus low and not used as a continuous regressor.

Why are Anovas used? Which are the multiple levels? Anova tests should be corrected for multiple comparisons (main effects and interactions), see
Cramer, A. O. J., van Ravenzwaaij, D., Matzke, D., Steingroever, H., Wetzels, R., Grasman, R. P. P. P., … Wagenmakers, E.-J. (2015). Hidden multiplicity in exploratory multiway ANOVA: Prevalence and remedies. Psychonomic Bulletin & Review, 23(2), 640–647. doi:10.3758/s13423-015-0913-5.

Since the results are obtained using scalp recordings, it cannot be concluded that the results are specific to the human brain, and to a specific location within it.

Fatigue and time on task are not taken into accound, and they have been proven to influence the spectral power in attention related tasks.



**Technical Rigor:**

2: Marginally convincing

---

### Official Review · AnonReviewer3 · 2019-09-26
**Useful application. Final conclusions need further support.**

**Clarity:** 4

**Comment:**

The authors establish a correlation between EEG-SSVEP power and perceptual accuracy in an attention task.

However, the final conclusions need further support. The result that accuracy increases for target but not distractor warrants further investigation. Is there a correlation between the target and distractor SSVEP power, and should we expect an effect because of this? The negative result on reaction time also needs further analysis: could this be because the high threshold was not sufficiently high?

**Category:**

Not applicable

**Clarity Comment:**

The submission is clear.

**Evaluation:**

3: Good

**Importance:**

3: Important

**Importance Comment:**

The authors present a useful application for the BMI: delivering a stimulus during different attention states.

**Intersection:**

2: Low

**Intersection Comment:**

Applicable to neuro.

**Rigor Comment:**

It would have been useful to do a thorough comparison of how different high/low threshold values could affect the results.

**Technical Rigor:**

3: Convincing

---

### Decision · Program_Chairs · 2019-10-01

**Decision:**

Reject

**Comment:**

Unfortunately, we had more submissions than we could accept and based on the review process, we have decided not to accept your submission.  Nevertheless, thank you for your submission and interest in our workshop.